# The association between community solidarity and adoption of public health preventive measures during the COVID-19 pandemic in a cross-sectional, multi-national sample

Jill Murphy[1]*, Michelle Sarah Livings[2], Martin Wong[3,4], Junjie Huang[3,4], Wanghong Xu[5], Andrés Caicedo[6,7], María Belen Arteaga[6,7], Harry H. X. Wang[8], Pramon Viwattanakulvanid[9], Erlinda C. Palaganas[10], Maria de Jesus Medina Arellano[11], Gil Soriano[12], Mellissa Withers[13]

**1** Department of Psychiatry, University of British Columbia, Vancouver, British Columbia, Canada, **2** School of Public & International Affairs, Princeton University, Princeton, New Jersey, United States of America, **3** The Faculty of Medicine, JC School of Public Health and Primary Care, The Chinese University of Hong Kong, Sha Tin, New Territories, Hong Kong, China, **4** The Faculty of Medicine, Centre for Health Education and Health Promotion, The Chinese University of Hong Kong, Sha Tin, New Territories, Hong Kong, China, **5** School of Public Health, Fudan University, Shanghai, China, **6** Universidad San Francisco de Quito USFQ, Colegio de Ciencias de la Salud, Escuela de Medicina, Quito, Ecuador, **7** Universidad San Francisco de Quito USFQ, Instituto de Investigaciones en Biomedicina iBioMed, Quito, Ecuador, **8** School of Public Health, The Sun Yat-sen University, Guangzhou, China, **9** College of Public Health Sciences, Chulalongkorn University, Bangkok, Thailand, **10** University of the Philippines, Baguio, Philippines, **11** Institute for Legal Research, National Autonomous University of Mexico, Mexico, Mexico, **12** Department of Nursing, College of Allied Health, National University, Manila, Philippines, **13** Keck School of Medicine, University of Southern California, Los Angeles, California United States of America,

* jkmurphy@stfx.ca

## Abstract

### Background

Few studies have examined the association between community solidarity and health-related behaviors. This study investigates solidarity in navigating challenges during the COVID-19 pandemic.

### Methods

We used cross-sectional data from a multi-national survey of 1,346 respondents to examine (1) factors relating to feelings of solidarity, and (2) associations between solidarity and public health preventive behaviors.

### Results

More than half (53.1%) of participants expressed feelings of solidarity; they were more likely to be aged 30 years or over, employed full-time, and residing in Eastern economies. We found a statistically significant association between positive feelings of solidarity and three of five COVID-19 prevention behaviors (social distancing,

which permits unrestricted use, distribution, and reproduction in any medium, provided the original author and source are credited.

**Data availability statement:** The data in this study are part of a larger dataset that is securely stored at the institution that led the multi-national survey (the Chinese University of Hong Kong). The informed consent process did not inform participants that their de-identified data would be made publicly available. Therefore, for ethical reasons we are unable to publicly share the dataset and data will be made available upon request to the authors. This study received primary ethics approval from the WHO Research Ethics Review Committee and Survey and Behavioural Research Ethics Committee (SBREC) of the Chinese University of Hong Kong. The SBREC may be contacted at: fssc02@cuhk.edu.hk

**Funding:** The author(s) received no specific funding for this work.

**Competing interests:** The authors have declared that no competing interests exist.

skipping an event, and masking in public). Those who reported previous influenza vaccination were also more likely to adopt these behaviors.

## Discussion

The findings underscore the potential of fostering community solidarity to enhance prosocial actions amid widespread emergencies.

---

## Background

The COVID-19 pandemic has had unprecedented health, social, and economic impact and has, in many settings, occurred alongside substantial social and political upheaval [1]. In the initial stages, health authorities across the globe implemented a range of public health interventions aimed at mitigating the virus's spread. These measures include social distancing, mask wearing, quarantine measures, closure of public spaces and businesses, and vaccination campaigns. The success of these measures largely relied on the willingness of citizens to adopt and cooperate in order to keep themselves and their communities safe. Just as these measures varied in their duration and intensity across countries and jurisdictions, the willingness of individuals and communities to adopt and sustain them has also varied.

In times of crisis, the propensity for communities to unite under a common cause is a well-observed phenomenon, emphasizing the crucial role of global collaboration and coordinated responses. Solidarity and social cohesion emerge as pivotal factors in deciphering the dynamics of individual and collective behavior in such periods. Solidarity can be defined as a series of actions leading from a sense of a shared experience and a desire to take care of others, particularly others who might be more vulnerable [2]. Feelings of solidarity may in turn lead to actions taken among individuals, groups, and institutions. For example, in the context of the pandemic, individual actions such as mask wearing and physical distancing were encouraged, while institutional actions included financial support for people experiencing job loss in some jurisdictions. Shönweitz et al. [2] described solidarity as a 'practice' with four characteristics. First, solidarity involves costs or direct action (e.g., financial, social, psychological), distinguishing it from strictly ideology or sentiment. Second, solidarity comes from shared experiences and contexts, which differentiates it from charitable acts. Third, solidarity is facilitated by a sense of reciprocity. Finally, solidarity is separate from acts resulting from friendship, love, or other profound relationships or feelings.

Strong levels of solidarity may in turn lead to increased social cohesion [3,4], which is crucial in crisis scenarios if local communities are to be able to implement efficient actions, promote community resilience and help overcome future challenges. Social cohesion has been defined in various ways, including as "the degree of social connectedness and solidarity between different community groups within a society, as well as the level of trust and connectedness between individuals and across community groups" [5]. Silveira et al. [6] note that despite the myriad definitions for the concept, there seems to be consensus that social cohesion is "an indicator of

togetherness in a society, and as such, it revolves around levels of interaction and integration, civic engagement and identity." A sense of belonging is identified as a key component of social cohesion, which involves feeling included within a specific context or environment [7].

The question of whether the adoption of pandemic precautions and other health behaviors (e.g., vaccination) is associated with social solidarity and cohesion has been examined as a means of understanding the variability in cooperation with and uptake of pandemic mitigation strategies. Liekefett et al. [8] found that concerns about personal protection and solidarity with at-risk groups contributed to this behavior during April and May of 2020 in Germany. Meanwhile, in a large international survey, Van Bavel et al. [9] found that a strong sense of national identity was associated with higher participation in public health measures and support for public health policies. A longitudinal qualitative study from Germany found that practices associated with feelings of solidarity decreased over time, and that participants' enthusiasm for these practices declined. At the same time participants maintained a sense of the importance of institutional measures to promote solidarity related not only to the direct health impacts of the pandemic, but also the worsening socioeconomic challenges faced by many, including seniors and youth [2].

Recent research has indicated that societies with a strong level of social cohesion have seen better outcomes during the COVID-19 pandemic [10–16]. Evidence suggests, for example, that both social cohesion and actions of solidarity have increased following events such as natural disasters, financial crises, and mass tragedies [17–20]. In socially cohesive communities, a stronger sense of "shared fate" means that individuals are more likely to make sacrifices or accept public health measures [13,14]. Similar to the findings related to solidarity, however, evidence also indicates that social cohesion is likely to increase during times of crisis but then diminish as the crisis continues [17,19,21,22]. Furthermore, the research findings indicate that investments in social cohesion yield dividends in the form of stronger, more interconnected communities. These enhanced social networks, in turn, are better equipped to navigate and respond effectively to crisis situations [20].

Though several studies show shifts in solidarity and related behaviors during the pandemic, there appears to be limited evidence regarding the sociodemographic factors associated with feelings of solidarity and social cohesion. One study [23] found that solidarity during the pandemic, as demonstrated by adherence to COVID-19 safety measures during the first and second lockdowns in Germany, was positively associated with adherence to safety measures and with being middle aged; solidarity was negatively associated with depression symptoms, male gender, and older age. However, to the best of our knowledge, there is limited research exploring these relationships on a global scale.

## Current study

This study examines solidarity and health behaviors during the COVID-19 pandemic. We used cross-sectional data from a global survey conducted throughout 2021 that captured respondents' perceptions of the pandemic and self-reported health behaviors. The primary aim of the study was to determine whether survey timing, sociodemographics, and/or health behaviors unrelated to COVID-19 were associated with feelings of solidarity during COVID-19. The secondary aim was to explore associations between solidarity and COVID-19-related health behaviors, such as self-reported social distancing, masking, and willingness to be vaccinated against COVID-19. The study was conducted in 11 economies (Canada, China, Ecuador, Hong Kong, Japan, Mexico, New Zealand, the Philippines, South Korea, Thailand, and the US) by an international team of more than 20 investigators that was formed through the Global Health Program of the Association of Pacific Rim Universities (APRU), a non-profit network of 60 universities from 19 economies of the Asia-Pacific. The APRU Global Health Program facilitates collaborative, multidisciplinary education, and research opportunities to address priority health concerns in the region.

## Data and methods

Data were obtained from a convenience sample via a global survey conducted by the APRU Global Health Program [24,25]. The survey was distributed online via social media and email. A panel of primary care providers, specialized

practitioners, and social epidemiologists completed pilot testing and validation of the survey. The survey was available in eight languages and included questions about demographics; socioeconomic status; health and wellbeing; experiences, behaviors, and perceptions related to COVID-19; and societal solidarity. Individuals were eligible to participate in the survey if they were 18 years of age or older; if they were capable of comprehending the purpose of the study; and if they provided informed consent. Individuals from 26 study sites responded to the survey between October 1st, 2020 and January 31st, 2022. Study sites included economies from the Asia-Pacific region [Australia, China, Hong Kong, India, Indonesia, Japan, Malaysia, New Zealand, the Philippines, Russia, South Korea, Taiwan, Thailand]; the Americas [Canada, Colombia, Ecuador, Mexico, Peru]; Europe [France, Germany, Italy, the United Kingdom]; and the Middle East [Iraq, Oman, Saudi Arabia]; note that not all study sites are represented in the current analysis due to the exclusion of participants with missing data. The Survey and Behavioral Research Ethics Committee of the Chinese University of Hong Kong (SBRE-20–035) approved the study, with ethical clearance obtained for all 26 study sites. The data were stored securely on an online platform and the database was password encrypted and accessible only by designated research personnel. Assuming that half of participants would report positive feelings of solidarity, we aimed to collect more than 1,111 validated surveys to achieve a precision level of more than 0.03.

A total of 2,713 individuals participated in the survey. About 59% of these individuals (n = 1,595) completed the solidarity scale–a key measure in this study, and thus a key inclusion criterion for the study sample. As this survey was administered to a cross-sectional convenience sample, advanced imputation methods were not feasible to preserve cases with missing data; thus, respondents with missing data on other measures of interest were excluded from the study sample. Of the 1,595 respondents who completed the solidarity scale, all respondents had complete data on the other outcomes of interest–the five COVID-19-related health behaviors. However, 12% were missing responses on other predictors of interest and were excluded from the study sample (n = 190 respondents were missing a country of residence, which prohibited us from determining whether they completed the survey before or after the COVID-19 vaccine was widely available in their country, and also limited us from coding their region of residence; n = 5 respondents did not report whether they had ever received a flu vaccine). An additional 54 respondents were missing data on one or more sociodemographic covariates and were excluded from the study sample. The study sample included 1,346 individuals with complete data on all outcomes, predictors of interest, and covariates, as described below.

## Measures

*Solidarity* was measured using a validated scale of three items [26] referring to feelings related to the COVID-19 pandemic in the past 14 days: (1) "I have felt there is greater solidarity and cohesion in our society and community," (2) "I have felt I am an integral part of our society or community," and (3) "I have felt our nation is growing closer together." Respondents rated their agreement with each statement on a Likert scale of 1 (not at all) to 5 (very much agree), and responses to the three items were averaged ($a$ = 0.80). A dichotomous indicator of solidarity was created by including positive scores in one category (average score greater than 3, coded 1), and combining negative and neutral scores into a second category (average score less than or equal to 3, coded 0).

Five *COVID-19-related health behaviors* were considered as outcomes of interest. The first four behaviors were measured with single items following the question stem: "During the last 14 days, which of the following measures have you taken to prevent infection from COVID-19?" (1) Social distancing was based on the item "Ensured physical distancing in public."(2) Staying home from work/school was based on the item "Stayed at home from work/school." (3) Skipping an event was based on the item "Avoided a social event I wanted to attend." (4) Masking *in public* was based on the item "Wore a mask in public." Respondents rated each item on a Likert scale of 1 (not at all) to 5 (very much). Responses greater than 3 were considered to endorse the health behavior (coded 1), and responses less than or equal to 3 were considered to *not* endorse the health behavior (coded 0). The fifth behavior, *willing to get the COVID-19 vaccine*, was based on a single question: for respondents completing the survey before a COVID-19 vaccine was available, "Are you

willing to receive the COVID-19 vaccine when one becomes available?" and for respondents completing the survey after a COVID-19 vaccine was available, "Are you willing to receive the COVID-19 vaccine?" Respondents rated their willingness on a Likert scale of 1 (strongly disagree) to 5 (strongly agree); responses greater than 3 were considered as willing to receive the vaccine (coded 1), and responses less than or equal to 3 were considered as *not* willing to receive the vaccine (coded 0). Note that these outcome measures were taken from Rek et al. [26] and the World Health Organization's tool for behavioral insights on COVID-19 [27].

*Covariates* were included to control for several potentially confounding factors. We controlled for survey timing in two ways: first, considering *month of survey completion*; and second, considering whether a respondent *completed the survey after the COVID-19 vaccine was made available to all adults in the respondent's country of residence* (yes = 1, no = 0). For the latter, internet searches were conducted to identify the date that a COVID-19 vaccine was available to all adults for each country represented in the analytic sample; estimated dates were used for several countries when exact dates of general availability could not be identified. Dates of COVID-19 vaccine availability for each study site represented in the analytic sample are included in S1 Tables.

Previous literature reveals that age, sex, education, and income are associated with practicing COVID-19 prevention measures, such as social distancing and wearing masks in public [28–38]. Thus, we additionally controlled for sociodemographics, including region of residence (East, including respondents residing in countries in the Asia-Pacific region except Australia, and West, including respondents residing in countries in the Americas, Europe, and Australia); age category (18–29, 30–39, 40–49, 50 and over); sex (male, female); education (less than 10 years, 10–12 years, more than 12 years); household composition (living alone, living with family, living with roommates, other); employment (full-time employed, part-time employed, self-employed, retired, student, not employed but not student, caregiver or other); financial situation in the past 6 months (decreased, stayed the same, increased, don't know); whether the respondent was receiving welfare at the time of survey completion (yes/no); and urban-rural status (rural area, rural-urban fringe, urban area). Lastly, we controlled for previous health behaviors that may predict feelings of solidarity and COVID-19-related health behaviors, specifically whether the respondent reported ever having a vaccine against influenza (yes/no).

## Analysis

First, we computed descriptive statistics for the full analytic sample, and disaggregated among respondents with positive feelings of solidarity and respondents with negative/neutral feelings of solidarity. Second, we conducted a logistic regression model predicting solidarity as a binary outcome ("Outcome 1"), including survey timing and covariates as predictors. Third, separate logistic regression models were specified to predict the five COVID-19-related health behaviors of interest ("Outcomes 2-6"). Three models were analyzed for each outcome: Model A was unadjusted and included solidarity as the only predictor; Model B was partially adjusted, including solidarity, availability of the COVID-19 vaccine at the time of survey completion, and whether respondent reported ever receiving a flu shot as predictors; and Model C was fully adjusted, including the three predictors from Model B as well as all covariates previously described.

## Results

### Descriptive statistics

Of the 1,346 respondents, just over half of the analytic sample reported positive feelings of solidarity (53.1%; *n* = 714). The majority of the sample endorsed each of the five COVID-19-related health behaviors of interest: 85.2% of respondents (*n* = 1,147) reported social distancing; 58.3% (*n* = 784) reported staying home from work/school; 59.7% (*n* = 804) reported skipping an event; 93.5% (*n* = 1,259) reported masking in public; and 80.6% of respondents (*n* = 1,085) were willing to get the COVID-19 vaccine. Close to two-thirds of respondents (61.5%; *n* = 828) completed the survey after the COVID-19 vaccine was widely available in their country of residence. Table 1 shows the descriptive statistics for the analytic sample, disaggregated by feelings of solidarity.

**Table 1. COVID-19-related health behaviors, survey timing, and sociodemographics of participants by feeling of solidarity.**

| | All respondents (N=1346) | Positive feelings of solidarity (n =714) | Negative/neutral feelings of solidarity (n =632) | p-value |
|---|---|---|---|---|
| Characteristic | n (%) | n (%) | n (%) | |
| COVID-19-related health behaviors: | | | | |
| Reported social distancing | 1147 (85.2) | 627 (87.8) | 520 (82.3) | **0.0043** |
| Reported staying home from work/school | 784 (58.3) | 374 (52.4) | 410 (64.9) | **< 0.001** |
| Reported skipping an event | 804 (59.7) | 426 (59.7) | 378 (59.8) | 0.9565 |
| Reported masking in public | 1259 (93.5) | 689 (96.5) | 570 (90.2) | **< 0.001** |
| Willing to get the COVID-19 vaccine | 1085 (80.6) | 576 (80.7) | 509 (80.5) | 0.9504 |
| Month of survey completion: | | | | **< 0.001** |
| January 2021 | 306 (22.7) | 224 (31.4) | 82 (13.0) | |
| February 2021 | 59 (4.4) | 37 (5.2) | 22 (3.5) | |
| March 2021 | 34 (2.5) | 16 (2.2) | 18 (2.9) | |
| April 2021 | 54 (4.0) | 37 (5.2) | 17 (2.7) | |
| May 2021 | 30 (2.2) | 19 (2.7) | 11 (1.7) | |
| June 2021 | 107 (8.0) | 59 (8.3) | 48 (7.6) | |
| July 2021 | 6 (0.5) | 4 (0.6) | 2 (0.3) | |
| August 2021 | 43 (3.2) | 17 (2.4) | 26 (4.1) | |
| September 2021 | 121 (9.0) | 57 (8.0) | 64 (10.1) | |
| October 2021 | 399 (29.6) | 164 (23.0) | 235 (37.2) | |
| November 2021 | 26 (1.9) | 13 (1.8) | 13 (2.1) | |
| December 2021 | 8 (0.6) | 2 (0.3) | 6 (1.0) | |
| January 2022 | 153 (11.4) | 65 (9.1) | 88 (13.9) | |
| Completed survey after COVID-19 vaccine was made available to all adults in respondent's country of residence | 828 (61.5) | 361 (50.6) | 467 (73.9) | **< 0.001** |
| Region of residence: | | | | **< 0.001** |
| East (Asia) | 987 (73.4) | 555 (77.8) | 432 (68.4) | |
| West (Americas, Australia, Europe) | 358 (26.6) | 158 (22.2) | 200 (31.6) | |
| Age category | | | | **0.0018** |
| 18–29 | 971 (72.1) | 489 (68.5) | 482 (76.3) | |
| 30–39 | 195 (14.5) | 111 (15.5) | 84 (13.3) | |
| 40–49 | 95 (7.1) | 54 (7.6) | 41 (6.5) | |
| 50 and over | 85 (6.3) | 60 (8.4) | 25 (3.9) | |
| Sex | | | | 0.6524 |
| Male | 447 (33.2) | 241 (33.8) | 206 (32.6) | |
| Female | 899 (66.8) | 473 (66.2) | 426 (67.4) | |
| Education | | | | 0.3151 |
| Less than 10 years | 47 (3.5) | 27 (3.8) | 20 (3.2) | |
| 10-12 years | 222 (16.5) | 108 (15.1) | 114 (18.0) | |
| More than 12 years | 1077 (80.0) | 579 (81.1) | 498 (78.8) | |
| Household composition | | | | **0.0054** |
| Living alone | 117 (8.7) | 66 (9.2) | 51 (8.1) | |
| Living with family | 1083 (80.4) | 552 (77.3) | 531 (84.0) | |
| Living with roommates | 133 (9.9) | 89 (12.5) | 44 (7.0) | |
| Other | 13 (1.0) | 7 (1.0) | 6 (0.9) | |
| Employment | | | | **< 0.001** |

*(Continued)*

**Table 1.** (Continued)

| | All respondents (N=1346) | Positive feelings of solidarity (n = 714) | Negative/neutral feelings of solidarity (n = 632) | *p*-value |
|---|---|---|---|---|
| Full-time employed | 506 (37.6) | 311 (43.6) | 195 (30.9) | |
| Part-time employed | 64 (4.8) | 27 (3.8) | 37 (5.8) | |
| Self-employed | 38 (2.8) | 14 (2.0) | 24 (3.8) | |
| Retired | 13 (1.0) | 12 (1.7) | 1 (0.2) | |
| Student | 668 (49.6) | 323 (45.2) | 345 (54.6) | |
| Not employed but not student | 30 (2.2) | 16 (2.2) | 14 (2.2) | |
| Caregiver or other | 27 (2.0) | 11 (1.5) | 16 (2.5) | |
| Financial situation in past 6 months | | | | 0.4621 |
| Decreased | 311 (23.1) | 159 (22.3) | 152 (24.0) | |
| Stayed the same | 695 (51.6) | 369 (51.7) | 326 (51.6) | |
| Increased | 246 (18.3) | 140 (19.6) | 106 (16.8) | |
| Don't know | 94 (7.0) | 46 (6.4) | 48 (7.6) | |
| Receiving welfare at the time of survey completion | 305 (22.7) | 175 (24.5) | 130 (20.6) | 0.0848 |
| Urban-rural status | | | | 0.1078 |
| Rural area | 374 (27.8) | 182 (25.5) | 192 (30.4) | |
| Rural-urban fringe | 114 (8.5) | 59 (8.3) | 55 (8.7) | |
| Urban area | 858 (63.7) | 473 (66.2) | 385 (60.9) | |
| Reported ever having a vaccination against influenza | 771 (57.3) | 403 (56.4) | 368 (58.2) | 0.5087 |

*Note.* N = 1346 participants who had complete data on outcome and all predictors; **bold font** indicates a significant difference in positive versus negative/neutral feelings of solidarity among respondents at p-value < 0.05, as indicated by chi-square tests.

Disaggregating by feelings of solidarity, more respondents who reported positive feelings of solidarity also reported social distancing (87.8%) and masking in public (96.5%) compared to those who reported negative/neutral feelings of solidarity (82.3% and 90.2%, respectively). More respondents who reported negative/neutral feelings of solidarity, compared to those with positive feelings of solidarity, reported staying home from work/school (64.9% versus 52.4%) and completing the survey after the COVID-19 vaccine was widely available in their country of residence (73.9% versus 50.6%). Respondents with positive versus negative/neutral feelings of solidarity also differed in terms of region of residence, age category, household composition, and employment, as detailed in Table 1.

## Predictors of positive feelings of solidarity

The logistic regression model predicting positive feelings of solidarity (versus negative/neutral feelings of solidarity; Outcome 1) showed significant associations with survey timing, region of residence, age category, and employment (see Table 2). Specifically, individuals who completed the survey after the COVID-19 vaccine was widely available in their country of residence had 0.5 times the odds (95% confidence interval [CI] 0.3, 0.8) of reporting positive feelings of solidarity compared to those who completed the survey before the COVID-19 vaccine was widely available. Individuals residing in the East (i.e., economies in the Asia-Pacific region, except Australia) had 2.0 times the odds (95% CI 1.3, 3.0) of reporting positive feelings of solidarity compared to individuals residing in the West (i.e., economies in the Americas, Europe, and Australia). Individuals between the ages of 30 and 39, between the ages of 40 and 49, and ages 50 and over had 1.5 times the odds (95% CI 1.0, 2.2), 1.7 times the odds (95% CI 1.0, 2.9), and 3.2 times the odds (95% CI 1.8, 5.7), respectively, of reporting positive feelings of solidarity compared to individuals between the ages of 18 and 29. Further, individuals who were part-time employed and self-employed had 0.5 times the odds (95% CI 0.3, 0.9) and

**Table 2. Results of logistic regression predicting positive feelings of solidarity (Outcome 1).**

| Predictor | Odds Ratio (95% CI) |
|---|---|
| Month of survey completion (ref = January 2022) | |
| January 2021 | **3.32 (1.24, 8.87)** |
| February 2021 | 2.40 (0.80, 7.25) |
| March 2021 | 1.02 (0.35, 2.97) |
| April 2021 | 2.98 (0.97, 9.13) |
| May 2021 | 2.74 (0.81, 9.30) |
| June 2021 | **3.56 (1.79, 7.11)** |
| July 2021 | 4.37 (0.62, 30.90) |
| August 2021 | 0.95 (0.34, 2.68) |
| September 2021 | 2.73 (1.41, 5.28) |
| October 2021 | 1.41 (0.90, 2.20) |
| November 2021 | 1.91 (0.80, 4.57) |
| December 2021 | 0.65 (0.12, 3.38) |
| Completed survey after COVID-19 vaccine was made available to all adults in respondent's country of residence | **0.48 (0.28, 0.82)** |
| Region of residence (ref = West) | |
| East | **1.95 (1.26, 3.04)** |
| Age category (ref = 18–29) | |
| 30–39 | **1.50 (1.00, 2.23)** |
| 40–49 | **1.74 (1.04, 2.92)** |
| 50 and over | **3.21 (1.80, 5.71)** |
| Sex (ref = Male) | |
| Female | 0.99 (0.77, 1.28) |
| Education (ref = Less than 10 years) | |
| 10-12 years | 1.58 (0.82, 3.02) |
| More than 12 years | 1.26 (0.90, 1.77) |
| Household composition (ref = Living alone) | |
| Living with family | 1.15 (0.75, 1.77) |
| Living with roommates | 1.62 (0.92, 2.84) |
| Other | 0.84 (0.23, 3.06) |
| Employment (ref = Full-time employed) | |
| Part-time employed | **0.50 (0.27, 0.90)** |
| Self-employed | **0.37 (0.18, 0.79)** |
| Retired | 4.26 (0.51, 35.44) |
| Student | 0.89 (0.63, 1.26) |
| Not employed but not student | 0.72 (0.32, 1.63) |
| Caregiver or other | 0.66 (0.28, 1.55) |
| Financial situation in past 6 months (ref = Decreased) | |
| Stayed the same | 0.80 (0.60, 1.08) |
| Increased | 0.79 (0.54, 1.16) |
| Don't know | 0.75 (0.46, 1.24) |
| Receiving welfare at the time of survey completion | 1.45 (0.97, 2.06) |
| Urban-rural status (ref = Rural area) | |
| Rural-urban fringe | 1.12 (0.70, 1.78) |
| Urban area | 0.98 (0.73, 1.32) |
| Reported ever having a vaccination against influenza | 1.07 (0.83, 1.39) |

*Note.* N = 1346 participants who had complete data on outcome and all predictors; **bold font** indicates a statistically significant odds ratio.

0.4 times the odds (95% CI 0.2, 0.8), respectively, of reporting positive feelings of solidarity compared to those who were full-time employed.

## Solidarity and survey timing as predictors of COVID-19-related health behaviors

Solidarity predicted three of the five COVID-19-related health behaviors of interest, summarized in Table 3. In the fully adjusted logistic regression model predicting social distancing (versus not social distancing; Outcome 2), individuals who reported positive feelings of solidarity had 2.1 times the odds (95% CI 1.5, 2.9) of also reporting social distancing compared to those who reported negative/neutral feelings of solidarity (full results of Outcome 2–Model C in S1 Tables). This significant association was robust across Model A (unadjusted model) and Model B (partially adjusted model) as well.

The next set of logistic regression models predicted staying home from work/school (versus not staying home from work/school; Outcome 3). Although solidarity was associated with reporting staying home from work/school in the unadjusted Model A (OR=0.6; 95% CI 0.5, 0.7), this association was not significant in Model B or Model C (full results of Outcome 3–Model C in S1 Tables).

In the fully adjusted logistic regression model predicting skipping an event one wanted to attend (versus not skipping an event; Outcome 4), individuals who reported positive feelings of solidarity had 1.4 times the odds (95% CI 1.1, 1.8) of reporting skipping an event compared to those who reported negative/neutral feelings of solidarity (full results of Outcome 4–Model C in S1 Tables). This significant association was robust across Model A (unadjusted model) and Model B (partially adjusted model). In Model C, individuals who reported ever having a vaccine against influenza had 1.4 times the odds (95% CI 1.1, 1.9) of reporting skipping an event compared to those who reported never having a vaccine against influenza; this association was also robust in the partially adjusted Model B.

In the fully adjusted logistic regression model predicting masking in public (versus not masking in public; Outcome 5), individuals who reported positive feelings of solidarity had 3.3 times the odds (95% CI 1.9, 5.7) of reporting masking in public compared to those who reported negative/neutral feelings of solidarity (full results of Outcome 5–Model C in S1 Tables). Again, this significant association was robust across Model A and Model B. Additionally, in Model C, individuals who reported ever having a vaccine against influenza had 2.2 times the odds (95% CI 1.2, 3.8) of reporting masking in public compared to those who reported never having a vaccine against influenza; this association was robust in Model B as well.

The final set of logistic regression models predicted willingness to get the COVID-19 vaccine (versus not willing to get the COVID-19 vaccine; Outcome 6). Solidarity was not associated with Outcome 6 in Model A, Model B, or Model C (full results of Outcome 6–Model C in S1 Tables). However, in Model C, individuals who completed the survey after the COVID-19 vaccine was widely available in their country of residence had 1.9 times the odds (95% CI 1.3, 2.6) of being willing to get the COVID-19 vaccine compared to those who completed the survey before the vaccine was widely available. Note that surveys administered after the COVID-19 vaccine was widely available did not ask whether individuals had already received the COVID-19 vaccine. Further, individuals who reported ever having a vaccine against influenza had 1.9 times the odds (95% CI 1.3, 2.6) of being willing to get the COVID-19 vaccine compared to those who reported never having a vaccine against influenza. These associations were significant in the partially adjusted Model B as well.

## Discussion

Our study findings shed light on feelings of community solidarity and how it related to adoption of public health preventive behaviors during the COVID-19 pandemic in a multi-national sample of respondents. First, our results demonstrated that about half of our respondents reported positive feelings of solidarity. Respondents with positive feelings of solidarity were more likely to be employed in a full-time position. We also found an association between positive feelings of solidarity and being aged 30 years or over. Research from Germany during the COVID-19 pandemic similarly showed a positive association between solidarity and middle age, but found lower solidarity among adults over 64 years. Kaup et al. [23] suggest

**Table 3. Results of logistic regression models predicting COVID-19-related health behaviors.**

| Predictor | Model A<br>Unadjusted Odds Ratio<br>(95% CI) | Model B<br>Partially Adjusted Odds Ratio<br>(95% CI) | Model C<br>Fully Adjusted Odds Ratio<br>(95% CI) |
|---|---|---|---|
| *Outcome 2: Reported social distancing* | | | |
| Positive feelings of solidarity (ref = Negative/Neutral) | **1.55 (1.15, 2.10)** | **1.83 (1.33, 2.52)** | **2.05 (1.46, 2.89)** |
| COVID-19 vaccine available[a] (ref = Vaccine not available) | – | **1.90 (1.39, 2.61)** | 1.07 (0.28, 4.14) |
| Ever vaccinated against influenza[b] (ref = Never vaccinated) | – | 1.12 (0.82, 1.52) | 1.20 (0.84, 1.70) |
| *Outcome 3: Reported staying home from work/school* | | | |
| Positive feelings of solidarity (ref = Negative/Neutral) | **0.60 (0.48, 0.74)** | 0.97 (0.75, 1.26) | 1.30 (0.97, 1.74) |
| COVID-19 vaccine available[a] (ref = Vaccine not available) | – | **9.78 (7.52, 12.74)** | 1.58 (0.63, 3.95) |
| Ever vaccinated against influenza[b] (ref = Never vaccinated) | – | 0.91 (0.71, 1.18) | 1.13 (0.83, 1.53) |
| *Outcome 4: Reported skipping an event* | | | |
| Positive feelings of solidarity (ref = Negative/Neutral) | **1.99 (1.09, 2.99)** | **1.32 (1.04, 1.67)** | **1.40 (1.08, 1.80)** |
| COVID-19 vaccine available[a] (ref = Vaccine not available) | – | **3.03 (2.39, 3.85)** | 1.56 (0.65, 3.75) |
| Ever vaccinated against influenza[b] (ref = Never vaccinated) | – | **1.29 (1.03, 1.62)** | **1.44 (1.11, 1.87)** |
| *Outcome 5: Reported masking in public* | | | |
| Positive feelings of solidarity (ref = Negative/Neutral) | **3.00 (1.86, 4.83)** | **2.86 (1.75, 4.67)** | **3.31 (1.94, 5.66)** |
| COVID-19 vaccine available[a] (ref = Vaccine not available) | – | 0.79 (0.48, 1.29) | 0.92 (0.16, 5.24) |
| Ever vaccinated against influenza[b] (ref = Never vaccinated) | – | **1.39 (0.89, 2.16)** | **2.16 (1.24, 3.76)** |
| *Outcome 6: Willing to get the COVID-19 vaccine* | | | |
| Positive feelings of solidarity (ref = Negative/Neutral) | 1.01 (0.77, 1.32) | 1.15 (0.87, 1.53) | 1.38 (0.99, 1.89) |
| COVID-19 vaccine available[a] (ref = Vaccine not available) | – | **1.64 (1.24, 2.18)** | **1.26 (1.08, 1.45)** |
| Ever vaccinated against influenza[b] (ref = Never vaccinated) | – | **1.78 (1.35, 2.34)** | **1.86 (1.34, 2.57)** |

*Note.*

[a]Respondent completed the survey after COVID-19 vaccine was made available to all adults in the respondent's country of residence;

[b]Respondent reported that they received a vaccine against influenza at least once prior to survey completion. N = 1346 participants who had complete data on outcome and all predictors; **bold font** indicates a statistically significant odds ratio. Full results from Model C can be found in S1 Tables.

that the association between middle age and positive feelings of solidarity could be explained by having high caretaking responsibilities, for both children and older parents, in middle age. Unlike the findings from Germany, older adults in our sample also demonstrated high levels of solidarity. It is possible that this is due to the strong collectivist and family orientation among participants from Eastern societies; we found that respondents from Eastern countries were also more likely to report positive feelings than those in Western countries. Interestingly, no association was found between positive feelings of solidarity and sex, education, household composition, financial situation, or urban versus rural residence.

We also found that the majority of our participants reported adoption of (or willingness to adopt) the five public health measures of interest that were intended to reduce COVID-19 exposure and transmission. This is in line with other research conducted with a variety of populations, which has found that a majority of respondents reported adopting COVID-19 preventive public health measures such as physical distancing, wearing a mask, being willing to participate in contact tracing programs, and implementing personal hygiene practices, including studies from China [39], Canada [40,41], Hong Kong [42], the US [28,41,43], South Korea [37,38], and Malaysia [44].

The timing of when a participant completed the survey was an important factor in predicting feelings of solidarity. We found that those who completed the survey later in the pandemic–after the COVID-19 vaccine was widely available in their country of residence–were less likely to report positive feelings of solidarity than those who completed the survey earlier–before the vaccine was widely available. This is consistent with other work on solidarity during crises, which has shown that feelings of solidarity often increase during the initial phase of a crisis, due to an initial sense of embarking upon a shared negative experience [22,45,46]. However, as occurred during the COVID-19 pandemic, these widespread feelings of solidarity are short-lived and often devolve into feelings of social fragmentation [17,20,21], as demonstrated by rises in racism [47,48] and mistrust in public health measures in many regions [20,49].

We also found that individuals residing in countries categorized as the East were more likely to report positive feelings of solidarity compared to individuals residing in countries categorized as the West. The differences in East versus West may be explained by cultural orientations towards collectivism (Eastern societies) or individualism (Western societies), which often shape beliefs, attitudes, norms, and values [50,51]. "Collectivism" is a term that refers to viewing people as interdependent, which leads to prosocial behaviors intended to benefit the community as a whole [50]. Western societies tend to be individualistic, favoring individual freedom, and thus decisions in these societies may be more driven by self-interest [52]. Several studies have examined individualism versus collectivism during the COVID-19 pandemic and have found a negative relationship between individualism and uptake of prosocial public health behaviors [35,52–60]. For example, Gelfand et al.'s [61] cross-national comparative study found that countries with "tight" cultures such as China, South Korea, and Singapore (in which social norms favor authority, social control, and collective welfare over individual rights) were more successful at implementing pandemic control measures and therefore had significantly lower cases and deaths than "looser societies," such as the US. Further, Vietnam successfully used messages promoting solidarity and social responsibility to encourage adoption of public health preventive behaviors during the COVID-19 pandemic, capitalizing on feelings of social cohesion in this collectivistic society [62].

Collectivism may be protective against COVID-19 because of its association with conformity to social norms relating to public health prevention behaviors. Previous research across various disciplines has found subjective norms and social pressure to be important factors in predicting COVID-19 public health behaviors (such as vaccination, wearing masks, or complying with stay-at-home orders) [41,59,63–65]. Studies have found higher compliance rates with such policies in settings where perceived public approval is high or where perceptions of peer pressure are strong [10,65–68]. Relatedly, our results indicate that feelings of solidarity were positively associated with adoption of preventive behaviors relating to COVID-19. Those who reported positive feelings of solidarity also reported higher levels of social distancing and masking in public. Furthermore, our results showed that positive feelings of solidarity predicted social distancing, skipping an event, and masking in public. These findings are consistent with a growing body of research demonstrating that feelings of community solidarity and social cohesion are associated with adoption of public health measures [10–13,15,16].

The association between feelings of solidarity and adherence to public health measures suggests that efforts to promote solidarity can be beneficial during widespread public health emergencies. In a study in Tamil Nadu, India, high levels of trust and confidence in public institutions, which is identified by Bottoni [69] as a dimension of social cohesion, led to higher degrees of cooperation with public health measures [70]. The authors recommended several strategies to promote public trust during times of crisis, including strong communication strategies that are transparent but measured, avoidance

of stigmatizing practices or language which may lead to reluctance to test, and promotion of community engagement strategies that foster a sense of ownership by communities [70]. A study among Orthodox Jewish communities in Antwerp, Belgium [71] found that although there were high levels of mistrust towards the public health system, engagement between health authorities and community leaders early in the pandemic helped these communities to adopt culturally appropriate approaches to prevention. These examples demonstrate the need for direct community engagement to promote feelings of trust and solidarity, particularly among isolated or marginalized communities.

Adopting public health preventive behaviors during the pandemic not only protected communities as a whole but also had benefits to individuals within those communities [46,65,72]. High levels of community solidarity are associated with individual resiliency, optimism, and coping in times of crisis [10–16]. Studies found better individual health outcomes during the pandemic in closer-knit neighborhoods [73–77]. For example, a study of Orthodox Jewish communities in the US, Belgium, and Israel found that a strong sense of social support, social identity, and feelings of belonging within these communities were associated with higher resilience, lower levels of psychological distress, and fewer negative personal impacts during the pandemic [71]. In Argentina, Carter and Cordero [10] found that optimism, resilient attitudes, and personal competence to face the pandemic were significantly higher among participants who perceived strong social ties in their neighborhoods. Further, those who perceived high neighborhood cohesion had more confidence in their neighbors' compliance with public health measures. In the UK, Lalot et al. [20] found that higher levels of social cohesion related to greater subjective well-being and optimism for the future.

We found that people who had previously adopted public health measures like the influenza vaccination were also more likely to adopt current COVID-19 pandemic measures, including skipping an event, masking in public, and being willing to get the COVID-19 vaccine. This finding aligns with recent studies demonstrating that previous influenza vaccination predicted COVID-19 prevention behaviors in Israel [78], South Korea [37], Italy [79], and the US [80–82]. Therefore, along with government mandates, public health campaigns that encourage the public to practice general preventive behaviors may be important in times of crisis.

## Limitations

This study has several limitations. First, it is a cross-sectional convenience sample and may not be generalizable to a larger population. The survey did not include questions about field of employment or work environment, and we were thus unable to control for field of employment (and, in particular, whether a respondent worked in health care or public health) in our regression models. Further, the online administration of the survey–and the recruitment of participants, specifically via social media–may have excluded those without immediate access to the internet, as well as those who were not active on social media. However, internet use is common in the countries sampled, and the sample of respondents gives us an important snapshot into feelings and behaviors during the pandemic. Second, we collected the data over about 15 months, and thus the local COVID-19 situation and personal knowledge and attitudes may have varied across different timepoints, but we controlled for survey timing in two ways in our regression models to account for these likely variations. Third, there are many ways to measure feelings of community solidarity; while we used a validated three-item measure, this measure did not include all possible dimensions of solidarity. Lastly, we coded solidarity as a dichotomous variable, and thus were unable to distinguish between "neutral" and "negative" feelings of solidarity, but this coding allowed us to focus on "positive" feelings of solidarity in a binary context.

## Conclusions

To conclude, our findings show that individuals reporting positive feelings of solidarity were more likely to adopt public health preventive measures during the COVID-19 pandemic. Governments and communities should take measures to foster solidarity and resilience among their residents by promoting prosocial behaviors in local neighborhoods and communities, strengthening community social support [52,62], and engaging with communities directly to ensure prevention

strategies are appropriate [70,71]. Promoting engagement in prosocial behaviors as a social norm could create an upward spiral for people to support each other during difficult times and could improve individuals' physical and mental health.

## NOTES

We use the term "economies" instead of "countries" to be consistent with APEC guidelines (APEC, 2024)

## Supporting information

**S1 Tables. S1 Table 1: Estimated dates of widespread COVID-19 vaccine availability for countries with residents included in study sample. S1 Table 2: Results of logistic regression predicting social distancing during COVID-19. S1 Table 3: Results of logistic regression predicting staying home from work/school during COVID-19. S1 Table 4: Results of logistic regression predicting skipping an event one wanted to attend during COVID-19. S1 Table 5: Results of logistic regression predicting masking in public during COVID-19. S1Table 6: Results of logistic regression predicting willingness to get the COVID-19 vaccine.**
(DOCX)

## Acknowledgments

The authors wish to thank the study participants for taking the time to complete this survey.

## Author contributions

**Conceptualization:** Martin Wong, Mellissa Withers.

**Formal analysis:** Jill K. Murphy, Michelle Sarah Livings, Junjie Huang, Mellissa Withers.

**Investigation:** Jill K. Murphy, Martin Wong, Wanghong Xu, Andrés Caicedo, Harry H.X. Wang, Pramon Viwattanakulvanid, Erlinda C. Palaganas, Maria de Jesus Medina Arellano, Gil Soriano, Mellissa Withers.

**Methodology:** Jill K. Murphy, Martin Wong, Junjie Huang, Mellissa Withers.

**Writing – original draft:** Jill K. Murphy, Michelle Sarah Livings, Mellissa Withers.

**Writing – review & editing:** Martin Wong, Junjie Huang, Wanghong Xu, Andrés Caicedo, María Belen Arteaga, Harry H.X. Wang, Pramon Viwattanakulvanid, Erlinda C. Palaganas, Maria de Jesus Medina Arellano, Gil Soriano.

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
