## [Decision Letter · Decision Letter 0]

Mar 15 2025

PONE-D-24-14173The association between community solidarity and adoption of public health preventive measures during the COVID-19 pandemic in a multi-national samplePLOS ONE

Dear Dr. Murphy,

Thank you for submitting your manuscript to PLOS ONE. After careful consideration, we feel that it has merit but does not fully meet PLOS ONE’s publication criteria as it currently stands. Therefore, we invite you to submit a revised version of the manuscript that addresses the points raised during the review process.

We look forward to receiving your revised manuscript.

Kind regards,

Jiankun Gong

Academic Editor

PLOS ONE

Additional Editor Comments:

The reviewers provided valuable feedback for revision. They recommend clearly stating that this is a cross-sectional study in the title and ensuring consistency in describing the study scope (e.g., "global survey" vs. "conducted in 11 economies"). Additionally, they suggest providing a more detailed explanation of statistical methods, including how confounders were controlled, how missing data were handled, and reporting confounder-adjusted estimates (e.g., 95% CI). Given differences in baseline characteristics (e.g., employment, household composition), please clarify how these were adjusted in the analysis. Reviewers also noted potential sample bias due to recruitment via social media and email and ask whether respondents working in healthcare were excluded. They suggest expanding on the connection between solidarity and COVID-19-related behaviors, adding the sample size (n) for key statistics, and further exploring the link between solidarity and age.

Regarding minor revisions, the manuscript is well-written but contains a small grammatical correction on page 4 ("a myriad of" should be "a myriad"). Reviewers also suggest enhancing clarity in the abstract to better reflect key findings. Please address these comments in your revision and let us know if further clarification is needed. We look forward to your revised manuscript.

Some papers should be consider to cite, because these papers closely examine solidarity during crisis.

Flynn, A. V. (2022). Solidarity and collectivism in the context of COVID-19. Nursing Ethics, 29(5), 1198-1208.

Gong, J., Zanuddin, H., Hou, W., & Xu, J. (2022). Media attention, dependency, self-efficacy, and prosocial behaviours during the outbreak of COVID-19: A constructive journalism perspective. Global Media and China, 7(1), 81-98. https://doi.org/10.1177/20594364211021331

Azimi, S., Andonova, Y. and Schewe, C. (2022), "Closer together or further apart? Values of hero generations Y and Z during crisis", Young Consumers, Vol. 23 No. 2, pp. 179-196. https://doi.org/10.1108/YC-03-2021-1300

Reviewers' comments:

Reviewer's Responses to Questions

**Comments to the Author**

1. Is the manuscript technically sound, and do the data support the conclusions?

Reviewer #1: Partly

Reviewer #2: Yes

2. Has the statistical analysis been performed appropriately and rigorously? 

Reviewer #1: No

Reviewer #2: Yes

3. Have the authors made all data underlying the findings in their manuscript fully available?

Reviewer #1: Yes

Reviewer #2: Yes

4. Is the manuscript presented in an intelligible fashion and written in standard English?

Reviewer #1: Yes

Reviewer #2: Yes

5. Review Comments to the Author

Reviewer #1: Thank you for giving me the opportunity to review this manuscript.

1) It is better to describe that this study was a cross-sectional study in the title.

2) Please describe all statistical methods to control for confounders. Please explain how missing data were addressed. It is important to describe unadjusted estimates and, if applicable, confounder-adjusted estimates and their precision (eg, 95% confidence interval). Please indicate number of participants with missing data for each variable of interest.

3) I think baseline characteristics were not the same between the two groups. In particular, there were differences on reported staying home from work/school, reported skipping an event, month of survey completion, region of residence, age category, household composition, employment. These factors are likely to modify the effects of the positive feelings of solidarity, and COVID-19-related health behaviors. Please describe how to control for these points in this study. I think it is necessary to describe confounder-adjusted estimates and their precision (the Model 2, and the Model 3, and so on).[

I think it is necessary to revise the manuscript.

Reviewer #2: This is a very interesting paper examining the association between community solidarity and health related behaviours. The COVID-19 pandemic and the expected behaviours have provided a very useful model in which to examine this relationship.

The authors have undertaken a useful piece of work that provides valuable information to inform public health decision makers in future scenarios.

There is a lack information however, about the sample characteristics in terms of factors that may influence response to the questionnaire. I would like to have known, for example, whether the authors specifically excluded respondents who were working in health care or public health. There is a high probability that the distribution of the survey via social media and email and within the circles of the authors would have recruited a sample of more well-informed and cautious social actors than the general public. Was there any exclusion of people with this potential bias?

You refer to the fact that this was a global survey (p.6) but you also state that it was “conducted in 11 economies”. These is a bit of a contradiction I think as it was technically conducted globally. Do you mean researchers were based in 11 economies? Clarify this please.

The finding that 4 out of the 5 measures of adherence to infection control guidelines were statistically significantly linked to levels of solidarity is a very interesting one. I think it could be made more clear in the abstract (p.2) precisely what you mean by this. It is clearer later in the paper but not in the abstract.

On page 10, where you give the % of positive feelings (51.1%) and the level of endorsement with the five COVID-19-related health behaviours, I think it would be helpful to give the n= as well.

The fact that levels of solidarity increased with the age (p.13) of the cohorts is one that I think would be of interest to explore a little more. Just a few more sentences.

The paper is of a very high editorial quality. There is just one error on page 4 in the introduction, you refer to “a myriad of”. Delete the word ‘of’ as this is redundant when using the word ‘myriad’.

A very interesting paper. Thank you for the opportunity to review.

6. PLOS authors have the option to publish the peer review history of their article (what does this mean? ). If published, this will include your full peer review and any attached files.

**Do you want your identity to be public for this peer review?** For information about this choice, including consent withdrawal, please see our Privacy Policy .

Reviewer #1: No

Reviewer #2: **Yes: ** Dr Angela V Flynn

---

## [Author Response · Author response to Decision Letter 1]

10 Apr 2025

April 3, 2025

Jiankun Gong

Academic Editor

PLOS ONE

Dear Jiankun Gong,

Thank you for the opportunity to review and resubmit our manuscript entitled “The association between community solidarity and adoption of public health preventive measures during the COVID-19 pandemic in a multi-national sample”. We have responded to the comments from the editor and reviewers below, with the reviewer comments in bold, and have highlighted the revisions in yellow in the updated manuscript.

Best wishes,

Dr. Jill Murphy

Research Chair in Mental Health and Addictions and Assistant Professor

Interdisciplinary Health Program

St. Francis Xavier University

Antigonish, Nova Scotia

Canada

1. Please ensure that your manuscript meets PLOS ONE’s style requirements, including those for file naming.

Response: We have made the following updates to adhere to PLOS ONE’s style requirements:

- We have updated the file name of the Supplementary Materials to adhere (S1Tables)

- We have added continuous line numbers in the manuscript document

- Please note that the affiliation of the first/corresponding author has changed since out original submission. We have highlighted this change in yellow on the cover page.

2. We note that you have indicated that there are restrictions to data sharing for this study. For studies involving human research participant data or other sensitive data, we encourage authors to share de-identified or anonymized data. However, when data cannot be publicly shared for ethical reasons, we allow authors to make their data sets available upon request. Please address the following prompts:

a. If there are ethical or legal restrictions on sharing a de-identified data set, please explain them in detail (e.g., data contain potentially identifying or sensitive patient information, data are owned by a third-party organization, etc.) and who has imposed them (e.g., a Research Ethics Committee or Institutional Review Board, etc.). Please also provide contact information for a data access committee, ethics committee, or other institutional body to which data requests may be sent.

b. If there are no restrictions, please upload the minimal anonymized data set necessary to replicate your study findings to a stable, public repository and provide us with the relevant URLs, DOIs, or accession numbers.

Response: The data in this study are part of a larger dataset that is securely stored at the institution that led the multi-national survey (the Chinese University of Hong Kong). The informed consent process did not inform participants that their de-identified data would be made publicly available. Therefore, for ethical reasons we are unable to publicly share the dataset and data will be made available upon request to the authors. This study received primary ethics approval from the WHO Research Ethics Review Committee and Survey and Behavioural Research Ethics Committee (SBREC) of the Chinese University of Hong Kong.

Additional Editor comments:

The reviewers provided valuable feedback for revision.

1. They recommend clearly stating that this is a cross-sectional study in the title and ensuring consistency in describing the study scope (e.g., “global survey” vs. “conducted in 11 economies”).

Response: We have added the term “cross-sectional” to the title. We feel that it is a global survey because it was conducted in 11 economies across the world. The title says “multi-national sample” which we feel adequately captures the study scope.

2. Additionally, they suggest providing a more detailed explanation of statistical methods, including how confounders were controlled, how missing data were handled, and reporting confounder-adjusted estimates (e.g., 95% CI). Given differences in baseline characteristics (e.g., employment, household composition), please clarify how these were adjusted in the analysis.

Response: We appreciate the reviewers’ questions and comments about the statistical methods utilized in this study. We took these comments as encouragement to add more detail to our Methods section, including a thorough description of how missing data were handled (p. 7-8) and the covariates that were included in adjusted logistic regression models (p. 9). We have also added results from unadjusted and partially adjusted regression models to complement the results from fully adjusted regression models in Table 3. We include 95% confidence intervals for all estimates in both Table 2 and Table 3. We feel that these new additions to the Methods section and tables, as well as additions to the Results section, have strengthened the presentation of the analysis and the manuscript overall, and we thank the Editor and the reviewers again for this feedback.

3. Reviewers also noted potential sample bias due to recruitment via social media and email and ask whether respondents working in healthcare were excluded.

Response: In response to Reviewer 2, Comment 2, we have added the following note on p. 23 in the Limitations section to acknowledge the limitations and potential biases related to the sampling strategy of this global survey: “The survey did not include questions about field of employment or work environment, and we were thus unable to control for field of employment (and, in particular, whether a respondent worked in health care or public health) in our regression models. Further, the online administration of the survey–and the recruitment of participants, specifically via social media–may have excluded those without immediate access to the internet, as well as those who were not active on social media.”

4. They suggest expanding on the connection between solidarity and COVID-19-related behaviors, adding the sample size (n) for key statistics, and further exploring the link between solidarity and age.

Response: In response to the reviewers’ comments, we have clarified the findings related to the relationship between solidarity and COVID-19 as described in detail in our response to Reviewer 1, Comment 3 on p. 4 below. We have added the sample size (n) for key statistics on p. 7-8 of the Methods section. We have also provided additional text discussing the relationship between age and feelings of solidarity in the Discussion on p. 18-19. Further details about these changes are described below on p. 5.

5. Regarding minor revisions, the manuscript is well-written but contains a small grammatical correction on page 4 (“a myriad of” should be “a myriad”).

Response: We have made this edit on p. 4.

6. Reviewers also suggest enhancing clarity in the abstract to better reflect key findings.

Response: In response to this comment we have added the following to the abstract on p. 2: We found a statistically significant association between positive feelings of solidarity and three of five COVID-19 prevention behaviors (social distancing, skipping an event, and masking in public).

7. Please address these comments in your revision and let us know if further clarification is needed. We look forward to your revised manuscript.

Response: Thank you!

8. Some papers should be considered to cite, because these papers closely examine solidarity during crisis:

Flynn, A.V. (2022). Solidarity and collectivism in the context of COVID-19. Nursing Ethics, 29(5), 1198-1208.

Gong, J., Zanuddin, H., Hou, W., & Xu, J. (2022). Media attention, dependency, self-efficacy, and prosocial behaviors during the outbreak of COVID-19: A constructive journalism perspective. Global Media and China, 7(1), 81-98.

Azimi, S., Andonova, Y., & Schewe, C. (2022). Closer together or further apart? Values of hero generations Y and Z during crisis. Young Consumers, 23(2), 179-196.

Response: Thank for your these suggestions. We have read the three recommended papers. We have added a reference to the Flynn (2022) paper on collectivism in the context of COVID-19 on p. 20.

Reviewers’ Comments:

Reviewer 1

1. It is better to describe that the study was a cross-sectional study in the title.

Response: Thank you for this suggestion. We have added this to the title.

2. Please describe all statistical methods to control for confounders. Please explain how missing data were addressed. It is important to describe unadjusted estimates and, if applicable, confounder-adjusted estimates and their precision (e.g., 95% confidence interval). Please indicate number of participants with missing data for each variable of interest.

Response: Thank you for this encouragement to provide more detail about our study sample and analysis. To address each of the points made in this comment:

We describe the covariates included in adjusted models on pp. 8-9. Models were adjusted for the following covariates: the month a respondent completed the survey; whether the survey was completed before or after the COVID-19 vaccine was made available to all adults in a respondent’s country of residence; respondent’s region of residence; respondent age; respondent sex; respondent educational attainment; respondent’s household composition; respondent employment status; respondent’s financial situation in the past 6 months; whether the respondent was receiving welfare at the time of survey completion; urban-rural status; and whether a respondent reported ever having a vaccine against influenza.

We present one set of estimates in Table 2, pertaining to “Outcome 1” (positive feelings of solidarity as the outcome of interest). The goal of this model was to explore what factors were significantly associated with positive feelings of solidarity; thus all covariates described above were included as predictors of interest. We have updated Table 3 to include three models for each of “Outcomes 2-6” (the five COVID-19-related health behaviors), described on p. 10: Model A is unadjusted and includes solidarity as the only predictor; Model B is partially adjusted, including solidarity, availability of the COVID-19 vaccine at the time of survey completion, and whether respondent ever received a flu shot as predictors; and Model C was fully adjusted, including the three predictors from Model B as well as covariates previously described. We include 95% confidence intervals for all estimates in both Table 2 and Table 3.

We have added clarification regarding how missing data were addressed. We analyzed data from a cross-sectional convenience sample and were thus unable to utilize advanced imputation methods to preserve cases with missing data. That said, we have added details, including the total number of individuals who participated in the survey, the number of participants with complete data on our outcomes of interest, the number of participants excluded due to missing data on predictors of interest, and the number of participants excluded due to missing data on covariates; these details are now included on pp. 7-8 of the Methods section.

3. I think baseline characteristics were not the same between the two groups. In particular, there were differences on reported staying home from work/school, reported skipping an event, month of survey completion, region of residence, age category, household composition, employment. These factors are likely to modify the effects of the positive feelings of solidarity, and COVID-19-related health behaviors. Please describe how to control for these points in this study. I think it is necessary to describe the confounder-adjusted estimates and their precision (Model 2, and Model 3, and so on).

Response: We acknowledge that some characteristics were significantly different between the two groups, as highlighted in the right hand column of Table 1. The first logistic regression model presented in the manuscript (“Outcome 1,” shown in Table 2) is intended to demonstrate which of these characteristics are significantly associated with positive feelings of solidarity. All variables described in the Methods section are included in the “Outcome 1” model in order to present adjusted estimates of the association between each characteristic and the “positive feelings of solidarity” outcome. Similarly, we include all variables described in the Methods section in Model C of the other logistic regression models (“Outcomes 2-6,” shown in Table 3) in order to present adjusted estimates of the associations between solidarity and each of these outcomes of interest. We have added unadjusted and partially adjusted estimates for “Outcomes 2-6” to Table 3; Model A for each outcome includes only solidarity as a predictor of interest, and Model B for each outcome includes solidarity, availability of the COVID-19 vaccine at the time of survey completion, and whether respondent ever received a flu shot. We include 95% confidence intervals for all estimates in both Table 2 and Table 3. We hope the inclusion of unadjusted and partially adjusted models to Table 3 helps to provide additional context to the adjusted models and to the manuscript as a whole.

Reviewer 2

1. This is a very interesting paper examining the association between community solidarity and health related behaviors. The COVID-19 pandemic and the expected behaviors have provided a very useful model in which to examine this relationship. The authors have undertaken a useful piece of work that provides valuable information to inform public health decision makers in future scenarios.

Response: Thank you for your positive feedback on this paper!

2. There is a lack of information, however, about the sample characteristics in terms of factors that may influence response to the questionnaire. I would like to have known, for example, whether the authors specifically excluded respondents who were working in health care or public health. There is a high probability that the distribution of the survey via social media and email and within the circles of the authors would have recruited a sample of more well-informed and cautious social actors than the general public. Was there any exclusion of people with this potential bias?

Response: We appreciate your concern about the sample characteristics. Unfortunately, the global survey did not include questions about employment field or work environment. Thus, we did not exclude participants who were working in health care or public health; however, we were also unable to include “field of employment” as a covariate in our adjusted regression models. We additionally acknowledge the potential bias associated with the sampling strategy, which involved recruiting survey participants via social media and email. On p. 23 in the Limitations section, we have added the following note: “The survey did not include questions about field of employment or work environment, and we were thus unable to control for field of employment (and, in particular, whether a respondent worked in health care or public health) in our regression models. Further, the online administration of the survey–and the recruitment of participants, specifically via social media–may have excluded those without immediate access to the internet, as well as those who were not active on social media.”

We additionally acknowledge your concern regarding the potential of other biases in this cross-sectional convenience sample. Indeed, the majority of the sample are under 30 years of age and have more than 12 years of education, and half of the sample are students, as shown in the descriptive statistics in Table 1. That said, we do include age, educational attainment, and employment status (including an option for “student”), along with a host of other factors that may influence an individual’s perspectives on solidarity as well as COVID-19-related health behaviors, as covariates in all adjusted regression models (see Tables 2 and 3).

3. You refer to the fact that this was a global survey (p. 6) but you also state that it was “conducted in 11 economies.” This is a bit of a contradiction I think as it was technically conducted globally. Do you mean researchers were based in 11 economies? Clarify this please.

Response: Thank you for this question. The research team was based in 20 economies but participants came from 11 economies throughout the world. So we feel that this is a global survey. H

---

## [Editor Report · Decision Letter 1]

The association between community solidarity and adoption of public health preventive measures during the COVID-19 pandemic in a multi-national sample

PONE-D-24-14173R1

Dear Dr. Murphy,

We’re pleased to inform you that your manuscript has been judged scientifically suitable for publication and will be formally accepted for publication once it meets all outstanding technical requirements.

Kind regards,

Jiankun Gong

Academic Editor

PLOS ONE
---

## [Editor Report · Acceptance letter]

PONE-D-24-14173R1

PLOS ONE

Dear Dr. Murphy,

I'm pleased to inform you that your manuscript has been deemed suitable for publication in PLOS ONE. Congratulations! Your manuscript is now being handed over to our production team.

Kind regards,

on behalf of

Dr. Jiankun Gong

Academic Editor

PLOS ONE